# Adapting waterpipe-specific pictorial health warning labels to the Tunisian context using a mixed method approach

Nadia Ben Mansour[1,2], Salsabil Rejaibi[1,2], Asma Sassi Mahfoudh[1], Sarra Ben Youssef[1], Habiba Ben Romdhane[1], Michael Schmidt[3], Kenneth D. Ward[4,5], Wasim Maziak[5,6], Taghrid Asfar[5,7,8]*

1 Research Laboratory on cardiovascular disease Epidemiology and Prevention, Faculty of Medicine of Tunis, University of Tunis El Manar, Tunis, Tunisia, 2 National Institute of Public Health, Tunisia, 3 Department of Art and School of Public Health, The University of Memphis, Memphis, Tennessee, United States of America, 4 School of Public Health, The University of Memphis, Memphis, Tennessee, United States of America, 5 Syrian Center for Tobacco Studies, Aleppo, Syria, 6 Department of Epidemiology, Robert Stempel College of Public Health, Florida International University, Miami, Florida, United States of America, 7 Department of Public Health Sciences, University of Miami, Miller School of Medicine, Miami, Florida, United States of America, 8 Sylvester Comprehensive Cancer Center, University of Miami, Miller School of Medicine, Miami, Florida, United States of America

* tasfar@miami.edu

## Abstract

### Background

Waterpipe (WP) use is rapidly increasing among young people worldwide due to the widespread misperception that it is safer than cigarette smoking. Health warning labels (HWLs) can effectively communicate tobacco-related health risks but have yet to be developed for WP. This study aimed to optimize and adapt a set of 16 pictorial WP-specific HWLs, developed by an international Delphi study, to the Tunisian context. HWLs were grouped into four themes: WP health risks, WP harm to others, WP-specific harms, and WP harm compared to cigarettes.

### Methods

Using a mixed method approach, we conducted ten focus groups combined with a survey among young WP users and nonusers (N = 63; age 18–34 years). In the survey, participants rated the HWLs on several communication outcomes (e.g., reaction, harm perception, effectiveness) and were then instructed to view all HWLs in each theme and rank them in the order of overall perceived effectiveness, from the most to the least effective. Afterward, participants provided in-depth feedback on HWLs and avenues for improvement. Mean effectiveness rating scores and percentages of participants' top-ranked HWLs were calculated. Discussions were audio-taped, transcribed verbatim, and analyzed thematically.

### Results

The top-ranked HWLs were those showing oral cancers, orally transmitted diseases, and ssa sick child. Focus group discussion illustrated that these selections were based on

**Funding:** This work was supported by the National Health Institute and Fogarty International Center (R01TW010654). The funders had no role in study design, data collection and analysis, decision to publish, or preparation of the manuscript.

**Competing interests:** The authors have declared that no competing interests exist.

participants' reactions to the direct impact of WP on a person's physical appearance and evoking guilt over children's exposure to WP smoke. Suggestions for improvement highlighted the need to use the local dialect and more affirmative statements (e.g., avoiding "may" or "can").

## Conclusions

This study is the first in North Africa to attempt to advance HWLs policy as the World Health Organization recommended. The results of this study can be used as a basis for implementing WP-specific health messages in the Eastern Mediterranean Region.

## Introduction

The Eastern Mediterranean Region (EMR) is recognized as a global hot spot for waterpipe (WP) use, especially among youth [1]. According to the World Health Organization (WHO) estimates in 2020, the current use of all tobacco products prevalence among men in Tunisia was 64.2%, the highest among all countries in the Middle East and North Africa (MENA) region, and it is rapidly increasing among women and adolescents [2, 3]. In particular, WP use was the highest (14%) among students aged 18–28 years old [4]. In addition, based on data from the Global Youth Tobacco Survey (GYTS) in Tunisia in 2017, the prevalence of current WP use increased from 5.8% in 2010 to 7.6% in 2017 among youth aged 13–15 years old [5, 6]. This increase is worrisome given accumulating evidence about the harmful effects of WP use on health, which in many aspects are similar to those of cigarettes (e.g., cancer, cardiovascular disease, respiratory disease, carbon monoxide poisoning) [7–9].

In response to the global tobacco epidemic, the WHO developed the Framework Convention on Tobacco Control (FCTC) to reduce the devastating health, social, environmental, and economic consequences of tobacco consumption by providing a framework for tobacco control measures to be implemented by its Parties at the national, regional and international levels [10]. Article 11 of the WHO FCTC requires the implementation of pictorial health warning labels (HWLs) on WP tobacco products [11, 12]. Pictorial HWLs have proven effective in communicating health risks associated with cigarettes and WP use [13, 14]. A pilot lab experiment study conducted by Maziak et al. among WP users in the US showed that placing a pictorial HWL on the WP device compared to no HWL (control) reduced users' positive experiences and exposure to exhaled carbon monoxide [15]. Tunisia has ratified the FCTC, but as in many other developing countries, HWLs policies are still not implemented in Tunisia [12, 16]. However, in February 2022, supported by WHO and the FCTC Secretariat, the Ministry of Health in Tunisia announced its intention to implement pictorial HWLs on the external packaging of tobacco products [17]. This positive development underscores the need for research to optimize HWLs design and content to the Tunisian context.

In 2017, our team started an international project aimed at developing and testing WP-specific pictorial HWLs for two countries in the EMR, Tunisia and Lebanon. As part of this project, we developed a set of 16 WP pictorial HWLs using the Delphi method with an international expert panel (Fig 1) [18]. These HWLs were grouped into 4 themes (T): T1) WP Health effects, T2) WP Harm to others, T3) WP-specific harms, and T4) WP harm compared to cigarettes. Using a mixed method approach that combined a rating survey with focus groups [18], the current study aimed to optimize and adapt these HWLs to the Tunisian context, focusing on young adults, who have the highest WP use prevalence in Tunisia. This study

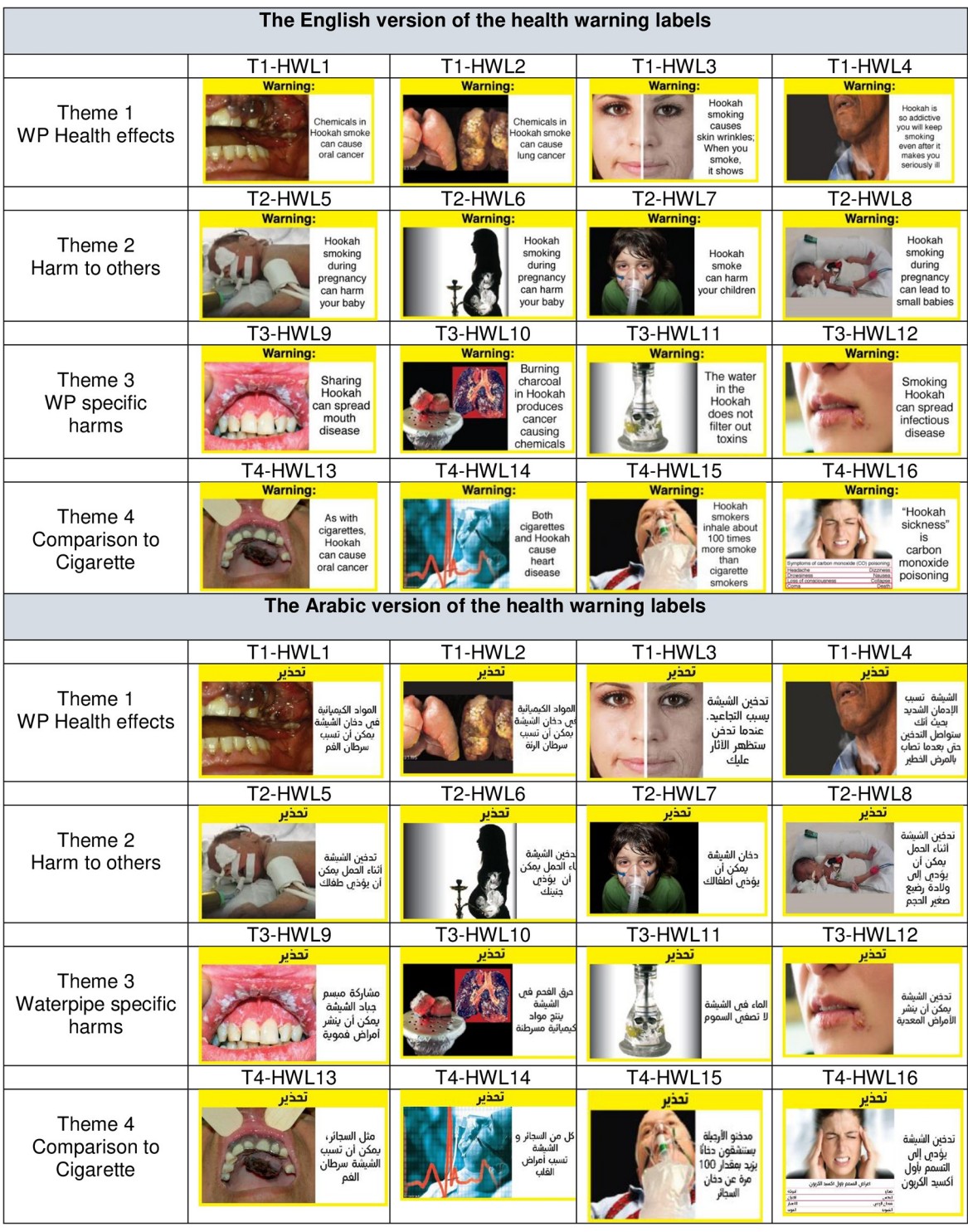

**Fig 1. Health warning labels tested in the study.**

reports on 1) young adults' rating and ranking of the overall effectiveness of the 16 HWLs, stratified by WP use status (WP users vs. nonusers) and gender (male vs. female), and 2) young adults' perceptions of HWLs' effectiveness regarding several communication outcomes (e.g., attraction, clarity, believability, relatedness, effectiveness) as elicited during focus group discussions. While the rating and ranking analysis will provide a snapshot of the evaluation of HWLs' effectiveness and help select and prioritize the top HWLs for further enhancement and development, focus group discussions will provide an in-depth understanding of participants' attitudes, feelings, beliefs, experiences, and reactions to the HWLs and identify salient suggestions for further HWLs adaptation and improvements. Results will provide Tunisia and other countries in the EMR with evidence-based HWLs that they can implement or further adapt and test within their specific context.

## Methods

### Study design

The Institutional Review Boards approved the study at the Faculty of Medicine of Tunis in Tunisia, Florida International University, and the University of Miami in the US. Using a mixed method approach [19, 20], we conducted 10 mixed-gender focus group discussions combined with a brief survey separately among current WP users (n = 23; defined as those who reported using WP at least once in the past 30 days) [21], and nonusers (n = 40; defined as those who have not used any tobacco/nicotine product in the past year) between January 2019 and January 2020. Participants received a $10 incentive for participating in the study.

### Participants and recruitment

A combination of online and offline recruitment methods was used: circulating recruitment postings on social media, university and student associations' websites, and distributing flyers at main university entrances. Additionally, snowball sampling was used by asking participants to refer their friends for the study. Prospective participants were screened for eligibility by phone, scheduled for a focus group session, and provided the time/place of their focus group. Eligibility requirements were being a young adult 18–34 years old and a current WP user or nonuser. This age group was selected because it is at high risk of prevalence, initiation, and progression of WP use [3]. In addition, given the important role of the HWLs regulations in preventing smoking initiation, including WP nonusers in the early stage of the HWLs development ensures maximizing their benefit for WP initiation prevention [22, 23]. Participants who reported using other tobacco products (e.g., cigarettes, e-cigarettes) were included to increase the generalizability of the results since concurrent tobacco use is common among young WP users in Tunisia [4].

### Procedures

Focus group sessions were held in a private conference room at the University of Tunis El Manar. Sessions lasted approximately 150 minutes and were conducted in Arabic, and they were moderated by the Tunisian project investigators and two public health graduate students trained in qualitative research. Each session started with a discussion on the nature and confidentiality of group discussions. Written informed consent was obtained, then participants completed a baseline assessment, including demographic characteristics and smoking history, followed by a brief survey to rate and rank the HWLs. Then, the focus group discussion started. Below we describe the methods of the brief survey and focus group discussion in more detail.

**Rating and ranking the HWLs.** In the brief survey, participants were instructed to view each label and rate it on a 5-point Likert-type scale (from 1 = strongly disagree to 5 = strongly agree) on four primary outcomes related to the HWLs' impact: 1) increase WP harm perception, 2) motivate users to quit, 3) prevent nonusers from initiating WP use, and 4) overall effectiveness on others [14, 24]. Participants were then instructed to view all HWLs in each theme and rank them in the order of overall perceived effectiveness, from the most to the least effective. To aid judgment of effectiveness, participants were asked to consider in their decision the effect of the HWLs in terms of attention (notice, general design); communication (clarity, understandability, believability); and effect (harm perception, intention to quit).

**Focus group discussion.** Focus group discussions entailed four 15- to 20-minute segments (one per theme). Each segment began with a PowerPoint presentation about the health effects of WP use relevant to the segment's theme, followed by a discussion of each HWL in that theme. The PowerPoint presentation was provided to generate basic and common knowledge about WP's harmful effects among participants. This knowledge can facilitate the group discussion around available evidence of WP health effects and related HWLs to which they will be exposed. Group discussions were guided by a semi-structured script based on the Message Impact Framework, which is based on communication [25, 26] and health behavior theories [27] and has been applied successfully in cigarette HWL research [14]. According to this model, focus group discussion focused on participants' perceptions of each HWL in terms of (1) attention (attraction, notice, engagement, general design); (2) emotional reaction (fear, believability, avoidance); (3) effect (harm perception, addictiveness perception, intention to quit, intention not to initiate WP use); (4) recommendation for improvement (relatedness to participants, message clarity, language level and synergy with pictorials); and (5) optimal placement (size and visibility of HWLs for each component—tobacco, device, charcoal) [14]. The discussion guide and the baseline assessment are provided in the supplement.

## Analysis

For the HWLs' rating and ranking analyses, the descriptive statistical analysis was conducted using SPSS software, version 21 (Version 21.0. Armonk, NY: IBM Corp) (Table 2). Analysis of the differences in the HWLs' ratings and rankings was stratified by WP use status (WP users vs. nonusers) and by gender (male vs. female) (Table 3). We stratified by gender because WP use prevalence, patterns, risk perceptions, and attitudes are different by gender in EMR, and because women have specific health concerns related to WP use effects on their offspring [28]. While "rating" indicates the mean index effectiveness, the "ranking" indicates the respondents' percentage ranking of the best HWL in each theme. Internal consistency of each HWL across the four outcomes of the rating assessment was conducted using Cronbach's alpha. The latter was demonstrated to be very high (Cronbach's alpha range 0.88–0.94) for all HWLs, so a mean effectiveness index was computed using the four measures [29]. The top-ranked HWL was identified for each theme by calculating percentages of participants' endorsement of HWLs in each theme. The Freidman test was used to compare effectiveness scores of HWLs per theme in each group, and Wilcoxon signed-rank test post hoc using Bonferroni correction for multiple comparisons (at $p < 0.05$) to examine pairwise significant differences between label effectiveness within each theme [24]. The Mann-Whitney U test was used to examine differences in HWL effectiveness ratings between WP users and WP nonusers and between females and males.

Focus groups were audio-recorded and transcribed verbatim and then translated into English in a 3-step quality control method comprising: 1) translating a completed translation back into Arabic, 2) comparing that new translation with the original text, and 3) reconciling

any meaningful differences between the two [30]. Data from focus group discussions were analyzed thematically using *Miner 4 Lite* software (QDA Miner 4 Lite, Provalis Research) [31]. Two research team members independently reviewed transcripts and developed a consensus plan to identify recurring themes and variants for pre-defined codes within the framework of qualitative content analysis [32].

The coding of participants was stratified based on WP use status (users, nonuser) and proceeded deductively from theoretical constructs used in the main categories (reaction, harm perception, intention to quit or not to initiate WP, and improvement). Meaning units (quotes) were grouped under their respective constructs (main categories), and summaries were drafted. A second coder reviewed the summaries and compared them to the data. Differences were resolved through peer debriefing [33]. The coding of participants' recommendations for HWL changes proceeded inductively, allowing for new main categories to emerge. The constant comparative method was used to identify patterns in the data and refine the categories [34]. Our frequent peer debriefing sessions included questioning each other's interpretations of responses from several perspectives: ethnic and social backgrounds, personal histories, and whether and to what extent our characteristics, prior experiences, and knowledge influenced our interpretations of the data. Where potentially biased interpretations arose, we assessed the fitness of meaning units (quotes) to their main categorization and preliminary codes [35–37]. This process was conducted iteratively as data analysis progressed.

## Results

### Participants' characteristics

Overall, 63 young adults participated in 10 focus groups (5 groups of users, 5 groups of nonusers). Among these, 50.8% were females and 36.5% were WP users. Nearly 47.8% of WP users perceived WP as less addictive than cigarettes, 56.5% reported using WP "just for fun," and 61% had no plan to quit WP use. Among nonusers, 46.3% reported that the main reason for not using WP is its harmful health effects, and 34.1% stated they were opposed to all types of smoking (Table 1).

### HWLs ratings

Top-rated HWLs within each theme were consistent across the four primary outcomes (harm perception, motivation to quit, preventing WP initiation, and overall effectiveness) (Table 2). The top-rated HWLs for the overall mean effectiveness score were T4-HWL13 "As with cigarettes, hookah can cause oral cancer" (Mean = 3.21 [38]) followed by T2-HWL5 "WP use during pregnancy can harm your baby" (2.92 [0.85]), T1-HWL1 "Chemicals in WP smoke can cause oral cancer" (2.90 [0.77]), and T1-HWL2 "Chemicals in WP smoke can cause lung cancer" (2.82 [0.76]) (Table 3). Only T4-HWL13 was significantly rated the highest within its theme (p = 0.02) (Table 3). In addition, males significantly rated three HWLs in Them 1 higher than females (T1-HWL1: oral cancer, p = 0.032; T1-HWL2: lung cancer, p = 0.010; T1-HWL4: the addictive nature of WP, p = 0.002). No statistically significant differences were found based on WP use status (Table 3).

### HWLs rankings

The top ranked HWLs for perceived overall effectiveness were T3-HWL9 "Sharing WP can spread mouth disease" (87.5%), followed by T4-HWL13 "As with cigarettes, WP can cause oral cancer" (60.0%), T1-HWL1 "Chemicals in WP smoke can cause oral cancer" (57.1%), and T2-HWL7 "WP smoke can harm your children" (42.1%) (Table 4).

**Table 1. Sociodemographic characteristics and waterpipe (WP) use attitudes and perceptions stratified by WP use status.**

| | All n = 63 | WP users n = 23 | Nonusers n = 40 |
|---|---|---|---|
| **Gender** | | | |
| Females | 32 (50.8) | 11 (34.4) | 21 (65.6) |
| Males | 31 (49.2) | 12 (38.7) | 19 (61.3) |
| **Age** | | | |
| 18–20 | 20 (31.7) | 11 (55.0) | 9 (45.0) |
| 21–25 | 18 (28.6) | 9 (50.0) | 9 (50.0) |
| 26–30 | 20 (31.7) | 3 (13.0) | 17 (42.5) |
| >30 | 5 (7.9) | - - - | 5 (12.5) |
| **Education** | | | |
| Undergraduate degree | 13 (20.6) | 6 (46.2) | 7 (53.8) |
| Graduate degree | 46 (73) | 16 (34.8) | 30 (65.2) |
| High school | 4 (6.3) | 1 (25.0) | 3 (75.0) |
| **Current cigarette smoker (yes)** | 28 (44.4) | 20 (71.4) | 8 (28.6) |
| **Perception of WP addiction compared to cigarettes** | | | |
| Less addictive | 18 (47.4) | 11 (50) | 7 (43.8) |
| Equally addictive | 9 (23.7) | 5 (22.7) | 4 (25) |
| More addictive | 9 (23.7) | 6 (27.3) | 3 (18.8) |
| Don't know | 2 (5.3) | - - - | 2 (12.5) |
| **Perception of WP harms compared to cigarettes** | | | |
| Less harmful | 2 (9.1) | 2 (11.1) | - - |
| Equally harmful | 6 (27.3) | 5 (27.8) | 1 (25) |
| More harmful | 11 (50) | 8 (44.4) | 3 (75) |
| Don't know | 3 (13.6) | 3 (16.7) | - - |
| **Reasons for not using WP*** | | | |
| Unhealthy | - - - | - - - | 19 (46.3) |
| Smells bad | - - - | - - - | 3 (7.3) |
| Makes me dizzy | - - - | - - - | 4 (9.8) |
| I am against WP use | - - - | - - - | 14 (34.1) |
| **Age of WP initiation*** | | | |
| 14–18 | - - - | 12 (54.5) | - - - |
| 19–25 | - - - | 7 (31.8) | - - - |
| 26–34 | - - - | 3 (13.6) | - - - |
| **Reason to use the WP*** | | | |
| Relax me | - - - | 4 (17.4) | - - - |
| Pass time | - - - | 13 (56.5) | - - - |
| My friends smoke | - - | 3 (13.0) | - - - |
| My family smoke | - - - | 1 (4.3) | - - - |
| **Own WP at home*** | | | |
| Yes | - - - | 7 (30.4) | - - - |
| No | - - - | 16 (69.9) | - - - |
| **Planning to quit WP use*** | | | |
| Within the next month | - - - | 4 (22.2) | - - - |
| Sometimes after 6 months | - - - | 5 (27.8) | - - - |
| Not planning to quit | - - - | 9 (50.0) | - - - |

* Not WP users;

** WP users

**Table 2. Health warning labels rating on harm perception, intention to quit waterpipe, intention to not initiate waterpipe use, and overall effectiveness stratified by themes and waterpipe use.**

| | Labels | Harm perception Mean (SD) | | | Intention to quit WP Mean (SD) | | | Intention to not initiate WP Mean (SD) | | | Overall effectiveness Mean (SD) | | |
|---|---|---|---|---|---|---|---|---|---|---|---|---|---|
| | | All | user | Nonuser | All | user | Nonuser | All | user | Nonuser | All | user | Nonuser |
| | | **Theme 1: WP Health Effects** | | | | | | | | | | | |
| T1-HWL1 | Chemicals in hookah can cause oral cancer. | 3.05 (0.80) | 2.81 (0.75) | 3.18 (0.81) | 2.92 (0.86) | 2.67 (0.85) | 3.05 (0.84) | 2.87 (1.02) | 2.71 (1.05) | 2.95 (1.01) | 2.79 (0.85) | 2.52 (0.92) | 2.93 (0.79) |
| T1-HWL2 | Chemicals in hookah can cause lung cancer. | 2.90 (0.79) | 2.82 (0.90) | 2.95 (0.72) | 2.87 (0.88) | 2.82 (0.90) | 2.90 (0.88) | 2.72 (1.00) | 2.64 (1.13) | 2.77 (0.93) | 2.80 (0.85) | 2.68 (0.94) | 2.87 (0.80) |
| T1-HWL3 | Hookah smoking causes skin wrinkles; When you smoke, it shows. | 2.54 (0.93) | 2.59 (0.90) | 2.51 (0.94) | 2.49 (0.94) | 2.55 (0.85) | 2.46 (0.99) | 2.41 (1.00) | 2.36 (1.00) | 2.44 (1.02) | 2.56 (0.90) | 2.59 (0.90) | 2.54 (0.91) |
| T1-HWL4 | Hookah is so addictive; you will keep smoking even after it makes you seriously ill. | 2.70 (0.92) | 2.64 (0.95) | 2.74 (0.92) | 2.67 (0.87) | 2.64 (0.84) | 2.68 (0.90) | 2.72 (0.97) | 2.91 (1.06) | 2.61 (0.91) | 2.68 (0.93) | 2.68 (1.04) | 2.68 (0.87) |
| | | **Theme 2: WP Harm to Others** | | | | | | | | | | | |
| T2-HWL5 | Hookah smoking during pregnancy can harm your baby. | 3.12 (0.93) | 3.00 (1.04) | 3.19 (0.85) | 2.98 (0.99) | 2.91 (1.04) | 3.03 (0.97) | 2.53 (1.05) | 2.57 (0.94) | 2.50 (1.13) | 3.07 (0.92) | 3.22 (0.95) | 2.97 (0.91) |
| T2-HWL6 | Hookah smoking during pregnancy can harm your baby. | 2.93 (0.81) | 2.90 (0.71) | 2.95 (0.86) | 2.83 (0.84) | 2.75 (0.71) | 2.87 (0.90) | 2.45 (1.02) | 2.35 (0.74) | 2.50 (1.15) | 2.83 (0.97) | 2.75 (0.96) | 2.87 (0.99) |
| T2-HWL7 | Hookah smoke can harm your children. | 3.05 (0.80) | 3.05 (0.72) | 3.05 (0.85) | 2.79 (0.91) | 2.64 (0.90) | 2.87 (0.92) | 2.43 (1.00) | 2.55 (0.96) | 2.36 (1.03) | 2.87 (0.86) | 2.77 (0.81) | 2.92 (0.90) |
| T2-HWL8 | Hookah smoking during pregnancy can lead to small babies. | 2.97 (0.82) | 2.91 (0.73) | 3.00 (0.88) | 2.93 (0.82) | 2.87 (0.62) | 2.97 (0.92) | 2.45 (1.08) | 2.48 (0.94) | 2.43 (1.16) | 2.83 (0.86) | 2.78 (0.85) | 2.86 (0.88) |
| | | **Theme 3: WP Specific Harms** | | | | | | | | | | | |
| T3-HWL9 | Sharing hookah can spread mouth disease. | 2.85 (0.94) | 2.68 (1.08) | 2.95 (0.85) | 2.66 (1.01) | 2.64 (1.09) | 2.67 (0.98) | 2.72 (1.08) | 2.55 (1.18) | 2.82 (1.02) | 2.77 (1.05) | 2.59 (1.22) | 2.87 (0.95) |
| T3-HWL10 | Burning charcoal in Hookah produces cancer causing chemicals. | 2.74 (0.94) | 2.65 (0.93) | 2.79 (0.96) | 2.70 (0.98) | 2.74 (1.01) | 2.68 (0.98) | 2.64 (1.03) | 2.52 (0.99) | 2.71 (1.06) | 2.62 (1.03) | 2.65 (1.07) | 2.61 (1.02) |
| T3-HWL11 | The water in the hookah does not filter out toxins. | 2.14 (0.98) | 2.10 (0.99) | 2.16 (0.98) | 1.98 (0.92) | 1.81 (0.87) | 2.08 (0.95) | 2.07 (1.02) | 1.86 (1.01) | 2.19 (1.02) | 1.98 (0.96) | 1.86 (1.01) | 2.05 (0.94) |
| T3-HWL12 | Smoking Hookah can spread infectious disease. | 2.45 (0.90) | 2.45 (0.94) | 2.45 (0.89) | 2.34 (0.84) | 2.35 (0.87) | 2.34 (0.84) | 2.40 (1.07) | 2.35 (1.13) | 2.42 (1.05) | 2.34 (0.90) | 2.25 (1.02) | 2.39 (0.85) |
| | | **Theme 4: WP Harm Compared to Cigarette** | | | | | | | | | | | |
| T4-HWL13 | As with cigarettes, hookah can cause oral cancer. | 3.36 (0.81) | 3.19 (0.92) | 3.46 (0.73) | 3.21 (0.81) | 3.19 (0.92) | 3.22 (0.75) | 3.07 (0.93) | 3.00 (1.04) | 3.11 (0.65) | 3.21 (0.81) | 3.1 (1.04) | 3.27 (0.65) |
| T4-HWL14 | Both cigarettes and hookah cause heart disease. | 2.88 (0.79) | 3.00 (0.79) | 2.82 (0.80) | 2.72 (0.83) | 2.85 (0.81) | 2.66 (0.84) | 2.79 (0.93) | 2.90 (0.91) | 2.74 (0.95) | 2.72 (0.91) | 2.60 (0.99) | 2.79 (0.87) |
| T4-HWL15 | Hookah smokers inhale about 100 times more smoke than cigarette smokers. | 2.90 (0.90) | 2.91 (0.81) | 2.89 (0.96) | 2.75 (0.95) | 2.82 (0.79) | 2.70 (1.05) | 2.63 (0.96) | 2.41 (0.85) | 2.76 (1.01) | 2.76 (1.04) | 2.59 (1.00) | 2.86 (1.05) |
| T4-HWL16 | Hookah sickness is carbon monoxide poisoning. | 2.99 (0.86) | 2.43 (0.87) | 2.79 (0.84) | 2.39 (0.83) | 2.19 (0.75) | 2.50 (0.86) | 2.46 (1.03) | 2.43 (1.07) | 2.47 (1.00) | 2.49 (0.91) | 2.29 (0.95) | 2.61 (0.88) |

*Participants were instructed to view each label and rate it on a 5-point Likert-type scale (from 1 = strongly disagree to 5 = strongly agree) for each outcome.

## Focus groups

Below we report the main results for each HWL according to the four themes.

**Theme 1—WP health effects.** *T1-HWL1*: *Chemicals in hookah can cause oral cancer.* Most participants found the text clear and provided "important information to know" (user). This HWL was found by most participants to provoke strong emotions and to be the most

**Table 3. Differences in the HWLs rating in terms of overall effectiveness stratified by gender and by waterpipe use status (users vs. nonusers).**

| | | Differences in effectiveness score by gender | | | Differences in effectiveness score by WP use status | | |
|---|---|---|---|---|---|---|---|
| | | All Mean (SD) | Male Mean (SD) | Female Mean (SD) | All Mean (SD) | Users Mean (SD) | Nonusers Mean (SD) |
| **Theme 1: WP Health Effects** | | | | | | | |
| T1-HWL1 | Chemicals in hookah can cause oral cancer. | **2.90 (0.77)**[b] | **3.15 (0.70)** | **2.67 (0.79)** | 2.90 (0.77) | 2.67 (0.73) | 3.02 (0.78) |
| T1-HWL2 | Chemicals in hookah can cause lung cancer. | **2.82 (0.76)**[b] | **3.09 (0.68)** | **2.58 (0.77)** | 2.82 (0.76) | 2.73 (0.84) | 2.87 (0.72) |
| T1-HWL3 | Hookah smoking causes skin wrinkles; When you smoke, it shows. | 2.50 (0.84) | 2.43 (0.95) | 2.56 (0.73) | 2.50 (0.84) | 2.52 (0.80) | 2.48 (0.87) |
| T1-HWL4 | Hookah is so addictive; you will keep smoking even after it makes you seriously ill. | **2.69 (0.84)**[b] | **3.02 (0.79)** | **2.36 (0.78)** | 2.69 (0.84) | 2.71 (0.85) | 2.67 (0.84) |
| **Theme 2: WP Harm to Others** | | | | | | | |
| T2-HWL5 | Hookah smoking during pregnancy can harm your baby. | 2.92 (0.85) | 1.03 (0.98) | 2.82 (0.69) | 2.92 (0.85) | 2.92 (0.88) | 2.92 (0.84) |
| T2-HWL6 | Hookah smoking during pregnancy can harm your baby. | 2.75 (0.82) | 2.83 (0.96) | 2.68 (0.69) | 2.75 (0.82) | 2.68 (0.63) | 2.79 (0.91) |
| T2-HWL7 | Hookah smoke can harm your children. | 2.79 (0.77) | 2.77 (0.92) | 2.80 (0.62) | 2.79 (0.77) | 2.75 (0.69) | 2.80 (0.82) |
| T2-HWL8 | Hookah smoking during pregnancy can lead to small babies. | 2.78 (0.79) | 2.93 (0.79) | 2.67 (0.79) | 2.78 (0.79) | 1.76 (0.63) | 2.81 (0.88) |
| **Theme 3: WP Specific Harms** | | | | | | | |
| T3-HWL9 | Sharing hookah can spread mouth disease. | 2.75 (0.94) | 2.77 (1.04) | 2.72 (0.86) | 2.75 (0.79) | 2.61 (1.06) | 2.82 (0.88) |
| T3-HWL10 | Burning charcoal in Hookah produces cancer causing chemicals. | 2.67 (0.92) | 2.89 (0.95) | 2.46 (0.85) | 2.67 (0.92) | 2.64 (0.88) | 2.69 (0.95) |
| T3-HWL11 | The water in the hookah does not filter out toxins. | 2.04 (0.88) | 2.25 (0.98) | 1.85 (0.75) | 2.04 (0.88) | 1.90 (0.85) | 2.12 (0.90) |
| T3-HWL12 | Smoking Hookah can spread infectious disease. | 2.75 (0.94) | 2.77 (1.04) | 2.72 (0.86) | 2.75 (0.94) | 2.61 (1.06) | 2.82 (0.77) |
| **Theme 4: WP Harm Compared to Cigarette** | | | | | | | |
| T4-HWL13 | As with cigarettes, hookah can cause oral cancer. | 3.21 (0.74) | 3.41 (0.56) | 3.00 (0.85) | 3.21[a] (0.74) | 3.11 (0.87) | 3.26 (0.66) |
| T4-HWL14 | Both cigarettes and hookah cause heart disease. | 2.78 (0.76) | 2.97 (0.80) | 2.59 (0.69) | 2.78 (0.76) | 2.83 (0.76) | 2.75 (0.77) |
| T4-HWL15 | Hookah smokers inhale about 100 times more smoke than cigarette smokers. | 2.75 (0.89) | 2.90 (0.96) | 2.62 (0.82) | 2.75 (0.89) | 2.68 (0.77) | 2.80 (0.96) |
| T4-HWL16 | Hookah sickness is carbon monoxide poisoning. | 2.50 (0.82) | 2.53 (0.89) | 2.46 (0.79) | 2.50 (0.82) | 2.33 (0.79) | 2.59 (0.83) |

[a] The overall effectiveness rating represents the mean score of four effectiveness domains including harm perception, intention to quit, preventing initiating WP use, and HWL's overall effectiveness on other people.

[b] p value <0.05 indicates a significant difference between the two comparison groups (males vs. females, or WP users vs. nonusers) after applying Mann Whitney test.

effective in this theme because it is "shocking, frightening and disgusting" (nonuser), and "it concerns the person's physical appearance, inspires people's curiosity and invites them to think about WP effects" (nonuser). However, some participants reported that because it is "too disgusting, people will avoid looking at the warning" (nonuser). Users, on the other hand, doubted the label's credibility and thought that "this could happen only to heavy WP users" (user).

*T1-HWL2*: *Chemicals in hookah can cause lung cancer.* This HWL was seen by most participants as outdated because "the text message is too much consumed, and the photo is very well known, so it is no longer attractive" (nonuser). users, in particular, felt that this label is ineffective because "it is specific to heavy users who are over 40 years, not for young WP users" and that "unconsciously, people would think that this could happen to others, but not for them".

*T1-HWL3*: *Hookah smoking causes skin wrinkles; When you smoke, it shows.* This HWL was judged by many participants as "not credible, because wrinkles may not be related to WP use" and that "the picture with make-up has diluted the value of the warning" (user). Besides, males

**Table 4. Health warning labels ranking in each theme ordered from the top to the least ranked (%).**

| Theme 1: WP Health Effects | 57.1 | 19.0 | 14.3 | 9.5 |
|---|---|---|---|---|
| | **T1-HWL1** Chemicals in hookah can cause oral cancer. | **T1-HWL2** Chemicals in hookah can cause lung cancer. | **T1-HWL4** Hookah is so addictive; you will keep smoking even after it makes you seriously ill. | **T1-HWL3** Hookah smoking causes skin wrinkles; When you smoke, it shows. |
| Theme 2: Harm to Others | 42.1 | 26.3 | 15.8 | 15.8 |
| | **T2-HWL7** Hookah smoke can harm your children. | **T2-HWL6** Hookah smoking during pregnancy can harm your baby. | **T2-HWL5** Hookah smoking during pregnancy can harm your baby. | **T2-HWL8** Hookah smoking during pregnancy can lead to small babies. |
| Theme 3: WP Specific Harms | 87.5 | 12.5 | - - - | - - - |
| | **T3-HWL9** Sharing hookah can spread mouth disease. | **T3-HWL11** The water in the hookah does not filter out toxins. | **T3-HWL12** Smoking Hookah can spread infectious disease. | **T3-HWL10** Burning charcoal in Hookah produces cancer causing chemicals. |
| Theme 4: Comparison to Cigarette | 60.0 | 20.0 | 13.3 | 6.7 |
| | **T4-HWL13** As with cigarettes, hookah can cause oral cancer. | **T4-HWL14** Both cigarettes and hookah cause heart disease. | **T4-HWL16** Hookah sickness is carbon monoxide poisoning. | **T4-HWL15** Hookah smokers inhale about 100 times more smoke than cigarette smokers. |

felt this HWL would not be practical because this side effect is "not important for males" (nonuser).

*T1-HWL4*: *Hookah is so addictive; you will keep smoking even after it makes you seriously ill.* The picture in this HWL, a man smoking through tracheotomy as an indicator of addiction, was seen as vague, "...unclear and not easy to understand" (nonuser). In addition, some participants with a medical education background doubted its credibility because it is "too much exaggeration, you cannot keep smoking at such an advanced stage of the disease!" (nonuser).

**Theme 2—WP Harm to others.** *T2-HWL5*: *Hookah smoking during pregnancy can harm your baby.* Although some participants found the picture of an intubated newborn baby "attractive and successful" and touching because "you feel that the baby is in a critical situation" (nonuser), others felt that this HWL is "very specific to pregnant women" (user). In addition, female participants found the HWL provocative and recommended: "avoid using baby's pictures to convey a health message" (user). Some participants also found that the text was not congruent with the picture because it was unclear: "How does maternal WP use affect the fetus in the picture?" (user).

*T2-HWL6*: *Hookah smoking during pregnancy can harm your baby.* This HWL was generally considered simple but ineffective in evoking emotions because "the message is purely scientific; it is not touching nor frightening" (nonuser). This explains why the effectiveness of this HWL was called into question: "It will not be effective in motivating a pregnant woman to quit WP use" (user).

*T2-HWL7*: *Hookah smoke can harm your children.* There was a consensus among participants that this HWL is very well designed and that "the black background is excellent, making it very attractive" (user). Guilt feelings were particularly stressed by nonuser females because "[they] felt sorry for the boy who is a victim of his own parents." On the other hand, some participants considered this HWL to lack believability: "The child seems sick more than being a victim of passive smoking" (nonuser). Overall, participants perceived this warning as not

effective, in the sense that the message received is to "avoid WP use around your child, but not quit WP use" (nonuser).

*T2-HWL8*: *Hookah smoking during pregnancy can lead to small babies*. Although the picture in this HWL was seen as "attractive and touching" (nonuser), participants considered this HWL to lack believability: "The photo could be related to any other pregnancy complications" (nonuser). The HWL was also considered ineffective because "low birth weight can be corrected easily" (nonuser).

**Theme 3—WP-specific harms.** *T3-HWL9*: *Sharing hookah can spread mouth disease*. The message was perceived as "new" and "important to know that sharing the mouthpiece is dangerous" (user). Participants (mainly user) found the photo the most attractive and repugnant and described its appearance as "shocking, disgusting and attractive." This HWL evoked fear and was considered ". . .very serious as it could happen even to light users" (nonuser). However, others perceived this HWL as ineffective and sending the wrong message by "discouraging sharing WP rather than discouraging WP use" because "the first thing that comes to mind is that changing the mouthpiece will solve the problem" (user). Nevertheless, many nonuser females were not familiar with the WP components and asked, "what does the word 'mouthpiece' mean?" (nonuser).

*T3-HWL10*: *Burning charcoal in Hookah produces cancer causing chemicals*. This HWL was seen as the least effective in this theme and sending the wrong message because the picture of the WP is attractive and very "good looking," to the extent that it was judged by one participant as "not effective in motivating people to stop WP use, and in contrast, it might encourage young people to try WP use" (user). Furthermore, the wording in the text was not clear and was "kind of making a scientific statement, which is hard to understand by ordinary people" (user).

*T3-HWL11*: *The water in the hookah does not filter out toxins*. Participants felt that the picture was unclear because "it is hard to distinguish the skull inside the bowl" (user). Several participants also mentioned that the picture "is attractive" and creates a favorable reaction to WP use. A user participant said, "the picture reminds me of a glass of lemonade." In addition, most participants were skeptical about the effectiveness of the HWL and thought that "this message won't be effective in making people stop WP use" (nonuser).

*T3-HWL12*: *Sharing hookah can spread mouth disease*. Participants were doubtful about the credibility and potential of this HWL because in their opinion, "the picture is not shocking, so it can't be attractive" (user) and "herpes is a benign and curable lesion and it can happen to anyone, regardless of WP use" (user). For that reason, this HWL was perceived as "not effective in encouraging people to stop WP use" (user).

**Theme 4—WP harm compared to cigarettes.** *T4-HWL13*: *As with cigarettes, hookah can cause oral cancer*. This HWL was considered "new, clear and provides important information," and that "comparing WP with cigarettes health effects is useful because most people generally underestimate the risk of WP use" (nonuser). According to a WP users participant, this HWL induced strong emotions—"shocking, but successful"—because "the picture is showing a disease that affects the physical appearance, which is more fearful than a picture of a damaged organ" (user). On the other hand, some participant felt that "the picture is very gruesome to the extent that people will try to avoid looking at the warning" (nonuser). One WP user was in denial and questioned the credibility of the HWL because "it is relevant only to heavy users so it will not make people stop WP use."

*T4-HWL14*: *Both cigarettes and hookah cause heart disease*. Although this HWL did not arouse emotions and was deemed as "not touching," and not informative because "it is very well known" (user), it was perceived as effective: "This warning presents a sudden heart attack . . .. it is a reminder that this can happen to anyone, especially those who are at risk for this

disease" (user). Another participant indicated that "the brutality of this event 'heart attack,' is frightening, unlike cancer, which have long latency period" (user).

*T4-HWL15*: *Hookah smokers inhale about 100 times more smoke than cigarette smokers.* This HWL was seen as not credible and an exaggeration: "Having such a huge difference in nicotine content between cigarettes and WP—100 times—decreases the warning credibility" (user). Participants also felt that "the picture is attractive and touching; however, it is not well connected to the text" (nonuser), which directly impacted their perceived effectiveness: ". . . So people will not be convinced to change their behaviors and stop WP use" (nonuser).

*T4-HWL16*: *Hookah sickness is carbon monoxide poisoning.* This HWL was found to lack believability: "Not credible! Headache does not reflect exactly CO intoxication" and "the picture does not really correspond to CO intoxication" (nonuser). Adding to this, a user indicated that "the message could be taken as advice to smoke in an open space to prevent exposure to CO rather than stopping WP use" (user).

**Participants' suggestions for HWLs improvement.**   Overall, most participants suggested that the message must be affirmative and should avoid the use of "may lead to" because it reduces the HWLs' effectiveness in communicating the harmful effects of WP use. Using the Tunisian dialect for the HWLs' text supported by scientific facts was also recommended for all HWLs to improve comprehension. They also recommended improving the clarity of the pictures. Some male participants suggested adding HWLs related to WP use's effect on men's sexual performance. Participants did not specify a preference for the best placement for the HWLs on WP components (tobacco, device, charcoal).

## Discussion

Tobacco research in Tunisia suffers from significant knowledge gaps. Despite the rise in WP use prevalence, research has not covered this important trend adequately and robustly. This study reports on the second phase of an international project aimed at helping Tunisia in developing and implementing WP-specific HWLs and provides the needed information to bridge this gap. Using a mixed-method approach, we involved young adults in a brief survey combined with focus group discussion to evaluate and adapt a battery of WP-specific pictorial HWLs to Tunisia's specific cultural context. Our results indicated that HWLs showing external health effects, such as oral cancer and orally transmitted diseases, and harmful effects of WP use on children were the top rated and ranked HWLs in terms of overall effectiveness. Focus group discussion results mirrored the rating and ranking results and helped further explain what elements of the HWLs were prominent in determining participants' perceptions of effectiveness the HWLs and how to improve them. Mainly, HWLs that provoked strong emotions such as fear, disgust, and guilt were perceived more effective. The least rated HWLs were those illustrating scientific facts without showing a specific disease. Compared to females, males were more receptive to HWLs in Theme 1 "WP health effect and addictive nature." Participants recommended improving the image's quality, using Tunisian dialect and assertive statements, and avoiding the use of complex words to improve the HWL readability.

In our study, males more than females rated three HWLs in T1 "WP health effect" as effective, including oral cancer, lung cancer, and the addictive nature of WP. This could be explained by the higher prevalence rate and more intensive pattern of WP use among males compared to females in the EMR [38]. Males suffer from WP-related adverse health effects and dependence more than females, and therefore these warning messages are more relevant to them [38, 39]. However, in contrast to results from Western countries [40], no differences in participants' reactions to the HWLs were detected based on their WP use status. It is important to mention here that fact that WP use is widespread in the Middle East and is socially

acceptable as part of the culture and history of the region. The high level of nonusers' exposure to the habit and social interaction between users and nonusers might create a shared perception of WP's health effects [28, 41].

Images of external bodily damage (e.g., oral cancer) were deemed more effective than images of internal damaged organs (e.g., lung cancer). These results may reflect the younger age of our study group and their concern about their physical appearance. The fear of social stigma due to the deterioration of body image is consistently used to further increase cigarette HWLs' effectiveness [42]. These findings mirror patterns in other countries concerning cigarette warnings [43, 44]. In addition, even though the use of children's images on HWLs was considered a kind of *"psychological manipulation"* by young women who are human rights activists, guilt feelings were mainly expressed when exposed to an HWL showing a child victim of an asthma attack: *"Children are victims of their own parents!"* This result is consistent with prior research in other countries in the EMR. A recent qualitative Jordanian study showed that a HWL displaying a child in distress was the most recalled within a set of 10 HWLs [45]. Another study among university students in three countries (Jordan, Egypt, and Palestine) showed that HWLs related to protecting children from exposure to WP smoke (*"Protect your children*: *Don't let them be exposed to WP smoke"*) was the most effective in motivating WP users to quit [46]. This research emphasizes using HWLs with children's pictures to target users in Arab countries in the EMR. Indeed, employing guilt is commonly used in social marketing [47] and would be further effective in the middle eastern cultural context, which remains collectivist and conservative in many aspects and prioritizes the family over the individual [48]. On the other hand, the lung cancer HWL was found by most participants as outdated due to its previous overuse on cigarette packages. This result suggests that wear-out effects of common HWL themes can take place and compromise HWLs' effectiveness. Indeed, several studies on cigarette HWLs demonstrated that regular rotation and innovation of HWL design could improve efficacy and salience [49, 50].

The results of the HWLs' ranking and rating based on overall perceived effectiveness echoed those from the focus group discussion. For example, the top-ranked HWLs (e.g., oral cancers, orally transmitted infections, and harm to children) were also perceived as the most effective during the focus group discussions, which validates and reaffirms the potential value of these HWLs in eliciting strong emotions (e.g., disgust, fear, and guilt). Similarly, HWLs that received the least effective rating (e.g. depicting CO poisoning) were also perceived as the least believable and effective during the focus group discussions. Participants indicated that these HWLs provoked poor reaction because they are didactic, illustrating scientific facts about vague chemicals (e.g., toxins) in WP or showing illnesses that are not dangerous and easy to treat, such as herpes. In contrast to this result, these HWLs were considered important and effective by tobacco control experts in our Delphi study (19), which highlights the importance of involving the target population in the development of HWLs [51].

We believe that the part of the discussion about ways to improve the HWLs was very informative for the adaptations we need to introduce in general, regardless of the HWL message and picture. For example, improving the image's quality was a general recommendation to improve HWLs' effectiveness. The tobacco firms seem to be aware of image quality for the effectiveness of HWLs since they have manipulated (by tinting, darkening, and fading) HWLs' image quality on cigarette packs in Pakistan [52]. Concerning the text message, focus group participants suggested using statements that are more assertive (e.g., "WP use causes") rather than "may lead to" or "can cause." However, these recommendations need to be balanced with the level of evidence concerning these associations so we do not make assertions beyond what the evidence suggests [53]. In addition, the use of the Tunisian dialect was frequently suggested during focus group discussions. Finally, participants recommended avoiding using complex

language that might hamper the message clarity, especially among vulnerable groups such as children [54]. Future research on developing new HWLs in the Middle East can benefit from using the "Arabic Readability Metric and Tool" to enhance the warning text readability [55, 56].

Our next research direction is to improve the HWLs' design and content, based on our focus group study, and to further test them in a lab experiment [57]. We will work with a public health designer to fine-tune the HWLs based on participants' feedback. The top-ranked four HWLs that were selected from each theme will then be produced in high quality for testing in a lab study [57], while the developed HWLs will be used to advocate for the adoption of HWLs policies and disseminate knowledge about WP harmful effects to people in Tunisia and other Arab countries [58]. In the final stage of our project, we will conduct a situation analysis of the local tobacco control policy environment in Tunisia to understand the local policy context, actors, as well as support for and barriers to HWLs policy implementation [59]. The situation analysis will involve two levels: 1) analysis of official documents related to national tobacco control policy to understand policy frameworks and processes, organizational structure, and stakeholders involved in tobacco control provision and implementation, and 2) semi-structured interviews with key informants including policymakers, café owners, WP users/nonusers, civil advocates, and the media. The situational analysis results will provide a clear roadmap for effective implementation of WP HWLs.

The strengths of this study stem from being guided by a theoretical model of message impact and its originality [14]. This is the first study in Tunisia and the North African region to develop a set of HWLs that are responsive to WP's unique harm and users' perceptions by using a participatory approach involving young adults. In addition, using a mixed method approach was essential to provide timely feedback, maximize resources and the reliability of results (triangulation analysis) [19, 20], and deepen our understanding of young adults' perceived effectiveness of the HWLs and how to improve their design [60]. Including both users and nonusers and males and females helped comprehensively consider prevention and cessation outcomes. However, as with all qualitative research, our findings should be interpreted with caution due to the non-representativeness of the sample studied. There was a different age distribution by WP use status (users vs. nonusers). However, given the qualitative nature of the study, the narrow-targeted age group (18–34 years), and that we used the same recruitment methods for both groups, we do not anticipate this imbalance will affect our results.

## Conclusions

The WHO-FCTC has recommended including pictorial HWLs in regulatory and policy action to protect the public [11, 61]. Tunisia remains way behind all other Arab countries on FCTC implementation. HWLs regulations for both cigarettes and WP have not yet been implemented [62]. The weakness of tobacco policies highlights the urgent need to strengthen tobacco control research in Tunisia to support and inform policymakers. This study reports on the second phase of an international project to develop and adapt WP-specific pictorial HWLs to the Tunisian context. Pictorial HWLs arousing strong emotions, with visible health consequences, and depicting harm to children represent a potential target for public health WP control efforts in Tunisia. In contrast, HWLs illustrating scientific facts or exposure to chemicals without tangible harm or disease communication were perceived to be the least effective. These results will inform the implementation of WP-specific HWLs within a comprehensive tobacco policy tailored to WP use specificity in Tunisia and other countries in the Eastern Mediterranean Region.

## Supporting information

**S1 Dataset.**
(XLSX)

**S2 Dataset.**
(XLS)

## Acknowledgments

**Declarations**: We declare that the work described here has not been published previously, that it is not under consideration for publication elsewhere, that its publication is approved by all authors and tacitly or explicitly by the responsible authorities where the work was carried out, and that, if accepted, it will not be published elsewhere in the same form, including electronically, in English or in any other language, without the written consent of the copyright-holder.

## Author Contributions

**Conceptualization:** Nadia Ben Mansour, Habiba Ben Romdhane, Taghrid Asfar.

**Data curation:** Nadia Ben Mansour, Salsabil Rejaibi, Asma Sassi Mahfoudh, Sarra Ben Youssef.

**Formal analysis:** Nadia Ben Mansour, Salsabil Rejaibi, Taghrid Asfar.

**Investigation:** Nadia Ben Mansour, Salsabil Rejaibi, Asma Sassi Mahfoudh, Sarra Ben Youssef, Habiba Ben Romdhane.

**Writing – original draft:** Nadia Ben Mansour, Taghrid Asfar.

**Writing – review & editing:** Salsabil Rejaibi, Asma Sassi Mahfoudh, Sarra Ben Youssef, Habiba Ben Romdhane, Michael Schmidt, Kenneth D. Ward, Wasim Maziak.

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
