## [Decision Letter · Decision Letter 0]

24 Aug 2022

PONE-D-22-10070Adapting waterpipe-specific pictorial health warning labels to the Tunisian context using mixed method approachPLOS ONE

Dear Dr. Asfar,

Thank you for submitting your manuscript to PLOS ONE. After careful consideration, we feel that it has merit but does not fully meet PLOS ONE’s publication criteria as it currently stands. Therefore, we invite you to submit a revised version of the manuscript that addresses the points raised during the review process.

The reviewers have raised a number of major concerns. They feel the manuscript should outline a clearly-defined research question, and they request improvements to the reporting of methodological aspects of the study. The reviewers also note concerns about the statistical analyses presented and request re-analyses be completed.

Could you please carefully revise the manuscript to address all comments raised?

We look forward to receiving your revised manuscript.

Kind regards,

Thomas Phillips, PhD

Staff Editor

PLOS ONE

Journal Requirements:

When submitting your revision, we need you to address these additional requirements. 1. Please ensure that your manuscript meets PLOS ONE's style requirements, including those for file naming. The PLOS ONE style templates can be found at https://journals.plos.org/plosone/s/file?id=wjVg/PLOSOne_formatting_sample_main_body.pdf and https://journals.plos.org/plosone/s/file?id=ba62/PLOSOne_formatting_sample_title_authors_affiliations.pdf 2. Thank you for stating the following in your Competing Interests section:   The manuscript co-authors and I wish to express that we have no financial or other relationships that might lead to a conflict of interest. All authors have participated in the conception of this report, assisted in revising the manuscript for important intellectual content, and provided final approval of the enclosed report.
 Please complete your Competing Interests on the online submission form to state any Competing Interests. If you have no competing interests, please state "The authors have declared that no competing interests exist.", as detailed online in our guide for authors at http://journals.plos.org/plosone/s/submit-now  This information should be included in your cover letter; we will change the online submission form on your behalf. 3. We note that you have indicated that data from this study are available upon request. PLOS only allows data to be available upon request if there are legal or ethical restrictions on sharing data publicly. For more information on unacceptable data access restrictions, please see http://journals.plos.org/plosone/s/data-availability#loc-unacceptable-data-access-restrictions.  In your revised cover letter, please address the following prompts: a) If there are ethical or legal restrictions on sharing a de-identified data set, please explain them in detail (e.g., data contain potentially sensitive information, data are owned by a third-party organization, etc.) and who has imposed them (e.g., an ethics committee). Please also provide contact information for a data access committee, ethics committee, or other institutional body to which data requests may be sent. b) If there are no restrictions, please upload the minimal anonymized data set necessary to replicate your study findings as either Supporting Information files or to a stable, public repository and provide us with the relevant URLs, DOIs, or accession numbers. For a list of acceptable repositories, please see http://journals.plos.org/plosone/s/data-availability#loc-recommended-repositories. We will update your Data Availability statement on your behalf to reflect the information you provide. 4. Your ethics statement should only appear in the Methods section of your manuscript. If your ethics statement is written in any section besides the Methods, please move it to the Methods section and delete it from any other section. Please ensure that your ethics statement is included in your manuscript, as the ethics statement entered into the online submission form will not be published alongside your manuscript.  5. We note that Figure 1 includes an image of a participant in the study. As per the PLOS ONE policy (http://journals.plos.org/plosone/s/submission-guidelines#loc-human-subjects-research) on papers that include identifying, or potentially identifying, information, the individual(s) or parent(s)/guardian(s) must be informed of the terms of the PLOS open-access (CC-BY) license and provide specific permission for publication of these details under the terms of this license. Please download the Consent Form for Publication in a PLOS Journal (http://journals.plos.org/plosone/s/file?id=8ce6/plos-consent-form-english.pdf). The signed consent form should not be submitted with the manuscript, but should be securely filed in the individual's case notes. Please amend the methods section and ethics statement of the manuscript to explicitly state that the patient/participant has provided consent for publication: “The individual in this manuscript has given written informed consent (as outlined in PLOS consent form) to publish these case details”.  If you are unable to obtain consent from the subject of the photograph, you will need to remove the figure and any other textual identifying information or case descriptions for this individual.

Reviewers' comments:

Reviewer's Responses to Questions

**Comments to the Author**

1. Is the manuscript technically sound, and do the data support the conclusions?

Reviewer #1: Partly

Reviewer #2: Yes

Reviewer #3: Yes

2. Has the statistical analysis been performed appropriately and rigorously? 

Reviewer #1: No

Reviewer #2: Yes

Reviewer #3: Yes

3. Have the authors made all data underlying the findings in their manuscript fully available?

Reviewer #1: No

Reviewer #2: Yes

Reviewer #3: Yes

4. Is the manuscript presented in an intelligible fashion and written in standard English?

Reviewer #1: No

Reviewer #2: Yes

Reviewer #3: Yes

5. Review Comments to the Author

Reviewer #1: Adapting waterpipe-specific pictorial health warning labels to the Tunisian context

using mixed method approach

Reviewed for PLOS ONE

Due: July 21, 2022

The article describes the first study conducted in Tunisia regarding the effectiveness of 16 graphic health warning labels previously developed by an international Delphi study. Using a combination of surveys followed by focus groups, WP smoking and nonsmoking participants (n = 63 total, aged 18-34 years) rated and ranked the HWLs based on their perceived harm, quit motivation, and likelihood to prevent WP smoking initiation. HWLs with highest effectiveness included those showing oral cancers, orally transmitted diseases, and a sick child. Subsequent focus group discussion indicated that health effects impacting a person’s physical appearance and guilt feelings over exposing children to WP smoke were critical factors in effectiveness ranking. Suggestions for improvement to the HWLs effectiveness include using more direct language in local dialect for the text, and rotating images to avoid overexposure. The study provides important information about graphic HWLs for waterpipe in the Tunisian context that can be leveraged for effective tobacco control policy designed to reduce prevalence of WP use in the EMR. The focus group results are presented very well – it is very interesting to see the participant feedback for each warning label. However, methodological detail and strong recommendations based on the evidence obtained for addressing the stubborn perception that WP is a less harmful/less addictive form of combustible tobacco smoking are missing. Tables 2 and 3 are quite confusing and require revision for better clarity.

Recommendation: revise and resubmit

General criticism

The manuscript should be reviewed by a scientific editor: for example, words like “bowel” when “bowl” is meant, need to be corrected. Saying a participant was “in denial” is not a scientific statement, but instead a judgmental one, and should be removed.

Introduction

• Has Tunisia adopted any warnings for cigarettes or other tobacco? The background mentions that they have not implemented much from the FCTC, but it is unclear whether warnings are on cigarettes or other tobacco. It is mentioned in the conclusions, but perhaps it should be in the introduction?

• The authors mentioned that the labels were tailored to the culture. Please provide more detail on how this was done.

• Consider including results of the Adetona review:

Adetona O, Mok S, Rajczyk J, Brinkman MC, Ferketich AK. The adverse health effects of waterpipe smoking in adolescents and young adults: A narrative review. Tobacco Induced Diseases. 2021;19.

• What is meant by a “communication domain”?

• Please explain why T4-HWL16 is in the Comparison to Cigarette theme (Theme 4) – it seems to fit better in the “WP Specific Harms” (Theme 3).

Methods

• Please provide a rationale for the age range you targeted.

• Please provide a rationale for including nonsmokers in the study design.

• Please provide a rationale for stratifying by gender?

• What was the rationale for giving a presentation on the health effects of WP smoking? In practice, such additional information will not be provided when one views a warning label.

• It is unclear what people were actually asked in the discussion. The methods talk about the framework, but how the constructs were tapped is unknown. Please provide more detail.

• Qualitative coding methods are rather vague. Please provide more detail on what the researchers coded for.

Results

• There was a different age distribution by WP smoking status (nonsmokers had many more in the two upper age categories). Any comparisons between the groups could be impacted by age. Please discuss the impact of this or if it can be controlled for.

• The title for Table 2 should indicate what is being presented in the cells. Overall, the tables need to be revised to improve clarity. It’s also hard to map the label numbers to what is being portrayed.

• Please edit Table 2 to include the rating scale (1=strongly disagree, 5 = strongly agree) so that the table can be understood when it stands alone.

• For Table 3, the N (%) doesn’t make sense: what is in the cell is not N (%) (unless the percentages are all less than 1%). The ranking % in the last column is not clear. Please better clarify what is being presented.

• Please provide a better explanation of what the double asterisks footnote means in Table 3.

• Use of the word “theme” is assigned to both the HWL themes (1-4) and the rating categories (1-3). This is confusing. Suggest exclusively using “category” for the ratings.

• For better clarity, please make the terminology for the themes used in Figure 1 and the text in pages 13-16 identical.

• What does this statement mean?

As for placement of HWLs, participants did not report noticeable locations.

Discussion

• The cited study for the following sentence is inappropriate because it is a secondary analysis of the GYTS focusing on students aged 13-15 years old. Please revise

This could be explained by the higher prevalence rate and more intensive pattern of WP use among males compared to females in the EMR [27].

• Please provide a reference to support these statements:

Males suffer from WP-related negative health effects and dependence more than females, and therefore these warning messages are more relevant to them.

• The logic of why more males rated the addictive nature of WP label as more effective isn’t clear because the reader does not know if the participants are aware of this health disparity; in fact the study data indicate that over half of the participants believe WP smoking is less addictive than cigarette smoking. Please refine.

• The studies cited regarding overexposure of HWLs are cigarette smoking studies; thus they do not demonstrate anything for waterpipe. Since overexposure and rotation are important concepts for HWLs, please revise this language to be a recommendation in the face of what is known about cigarette HWLs.

• What is meant by “advanced language”? This may be a good place to cite the Flesch Readability scale to make your point.

Reviewer #2: This is a mixed methods design that examines how waterpipe tobacco smokers and nonsmokers rated and commented on several health warnings previously developed by the team. Overall, the strengths of this paper include: identifying further warnings that may ultimately be used in a specific geographic location; use of smokers and nonsmokers; providing insightful comments from the study participants both in terms of weaknesses, strengths, and ways to improve warning in various domains, and perhaps most importantly, showing how expert opinions diverge from the intended audience. With the said there are a few issues the authors may which to address. I note these below.

1. Would be useful to provide as supplemental materials the semi-structured script.

2. Any conjecture as to why smokers and nonsmokers do not differ in their evaluations. Also, it would be useful to state why we should focus on nonsmokers (e,g., to prevent uptake and/or susceptibility to use).

3. Describe future plans given that feedback was given as to how to improve the warnings.

4. Methodologically, the warnings were obtained using the Delphi approach. This seems reasonable. However, given the large discrepancies obtained, one wonders if there is another approach that may allow investigators to match what the target audience thinks more closely. For example, one suggestion is the mental models approach. Perhaps this can be added in the discussion section for ways to improve the development of hookah warnings.

Reviewer #3: In general, I found this to be a well-written paper on a relevant topic, that could have policy utility in Tunisia as well as in other countries with similar product use. I have a few minor suggestions for the text, and my main comment is that the specific warnings supported by this research, and the implications of this, should be made more explicit in the discussion and conclusions.

Minor points:

Would it be better to talk about water pipe users than smokers? This was somewhat confusing to me.

Is it possible to use more descriptive labels in tables 2 and 3 in order to make them more interpretable for the reader? If not, can the data be summarized and move the tables to appendix materials?

Its not clear how the stratification of focus group participants was used for comparison purposes in the focus group analysis.

Is there a way to more explicitly integrate findings from the rankings and the focus group discussion? In the discussion, it is mentioned that the rankings were confirmed during focus group discussions, but it is not clear how this assessment was made.

Findings from the focus groups are mentioned in the discussion that are not explored in the results section. The discussion should be considering interpretation of findings rather than a presentation of new information. How was the theoretical model called upon in the discussion?

Line 386, I believe that the authors meant qualitative – but qualitative research is usually not intended to be representative, as extrapolation from qualitative findings is not intended to be of this type.

In the conclusion section, it would be good to see what messages/HWL were supported by this research.

6. PLOS authors have the option to publish the peer review history of their article (what does this mean?). If published, this will include your full peer review and any attached files.

Reviewer #1: No

Reviewer #2: **Yes: **Isaac M. Lipkus

Reviewer #3: No

---

## [Author Response · Author response to Decision Letter 0]

18 Oct 2022

Thank you for giving us the opportunity to submit a revised draft of our manuscript entitled “Adapting waterpipe-specific pictorial health warning labels to the Tunisian context using mixed method approach” for publication in PLOS ONE. We appreciate the time and effort that you and the reviewers have dedicated to providing your valuable feedback on our manuscript. 

We provided a detailed point-by-point response to reviewers' comments at the bottom of this letter. Changes in the manuscript in response to reviewers' comments are highlighted in the marked copy of the manuscript.

We look forward to hearing from you in due time regarding our submission and to responding to any further questions and comments you may have. 

Response to reviewers and editor

Reviewer 1

Introduction

1. Has Tunisia adopted any warnings for cigarettes or other tobacco? The background mentions that they have not implemented much from the FCTC, but it is unclear whether warnings are on cigarettes or other tobacco. It is mentioned in the conclusions, but perhaps it should be in the introduction.

Response: Tunisia has ratified the FCTC, but as in many other developing countries, HWLs policies are still not implemented in Tunisia (12, 16). However, in February 2022, supported by WHO and the FCTC Secretariat, the Ministry of Health in Tunisia intended to implement pictorial HWLs on the external packaging of tobacco products. We modified paragraph 2 in the introduction to verify this information. It reads now as: 

“In response to the global tobacco epidemic, the WHO developed the Framework Convention on Tobacco Control (FCTC) to reduce the devastating health, social, environmental, and economic consequences of tobacco consumption by providing a framework for tobacco control measures to be implemented by its Parties at the national, regional and international levels (11). Article 11 of the WHO FCTC requires the implementation of pictorial health warning labels (HWLs) on WP tobacco products (12, 13). Pictorial HWLs have proven effective in communicating health risks associated with cigarettes and WP smoking (14, 15). A pilot lab experiment study conducted by Maziak et al. among WP smokers in the US showed that placing a pictorial HWL on the WP device compared to no HWL (control) reduced smokers’ positive experiences and exposure to exhaled carbon monoxide (16). Tunisia has ratified the FCTC, but as in many other developing countries, HWLs policies are still not implemented in Tunisia (13, 17). However, in February 2022, supported by WHO and the FCTC Secretariat, the Ministry of Health in Tunisia announced its intention to implement pictorial HWLs on the external packaging of tobacco products (18). This positive development underscores the need for research to optimize HWLs design and content to the Tunisian context.”

2. The authors mentioned that the labels were tailored to the culture. Please provide more detail on how this was done.

Response: The original labels developed in the Delphi study were not culturally adapted to the Tunisian context. The main purpose of the focus group study was to further optimize and adapt these labels to the Tunisian context based on the feedback of Tunisians. To clarify this, we added this information in the last paragraph of the introduction: 

“Using a mixed method approach that combined a rating survey with focus groups (19), the current study aimed to optimize and adapt these HWLs to the Tunisian context, focusing on young adults, who have the highest WP smoking prevalence in Tunisia.”

3. Consider including results of the Adetona review: Adetona O, Mok S, Rajczyk J, Brinkman MC, Ferketich AK. The adverse health effects of waterpipe smoking in adolescents and young adults: A narrative review. Tobacco Induced Diseases. 2021;19.

Response: We added this information and cited Adetona et al (reference # 10) at the end of the first paragraph of the introduction:

“This increase is worrisome given accumulating evidence about the harmful effects of WP smoking on health, which in many aspects are similar to those of cigarettes (e.g., cancer, cardiovascular disease, respiratory disease, carbon monoxide poisoning) (8-10).”

4. What is meant by a “communication domain”?

Response: We apologize for the confusion. The “communication domains” referred to participants’ attitudes, feelings, beliefs, experiences, and reactions to the HWLs. We added this information for more clarification. The sentence reads now:

“While the rating and ranking analysis will provide a snapshot of the evaluation of HWLs’ effectiveness and help select and prioritize the top HWLs for further enhancement and development, focus group discussions will provide an in-depth understanding of participants’ attitudes, feelings, beliefs, experiences, and reactions to the HWLs and identify salient suggestions for further HWLs adaptation and improvements.”

5. Please explain why T4-HWL16 is in the Comparison to Cigarette theme (Theme 4) – it seems to fit better in the “WP Specific Harms” (Theme 3).

Response: T4-HWL16 “Hookah sickness is carbon monoxide poisoning” was listed in the WP-specific harm theme since it is related mostly to charcoal emissions which is unique to this tobacco use method and produces level of CO exposure not seen in other combustion tobacco products (Bhatnagar, A., et al., Water pipe (hookah) smoking and cardiovascular disease risk: a scientific statement from the American Heart Association. Circulation, 2019. 139(19): p. e917-e936; Asfar, T., et al., Delphi study among international expert panel to develop waterpipe-specific health warning labels. Tobacco Control, 2020. 29(2): p. 159-167). 

Methods

6. Please provide a rationale for the age range you targeted. 

Response: The focus on young adults was based on two factors. First, WP smoking prevalence in Tunisia is the highest among young adults (based on the WHO FCTC, 2020 Report - Core Questionnaire of the Reporting Instrument of WHO FCTC). Second, this age group is particularly at-risk in terms of WP initiation. According to the Tunisian Health examination survey, the average age of initiation in Tunisia is 18.5. We added this information in the “Participants and Recruitment” section to clarify our rationale: 

“This age group was selected because it is at high risk of prevalence, initiation, and progression of WP use (4).” 

7. Please provide a rationale for including nonsmokers in the study design.

Response: HWLs policies play an important role in reducing smoking rates by de-normalizing smoking and preventing smoking initiation in addition to encouraging quitting (Hammond, D., et al., Showing leads to doing: graphic cigarette warning labels are an effective public health policy. The European Journal of Public Health, 2006. 16(2): p. 223-224). Almost 60% of youth in Canada, the United Kingdom, and Australia reported that HWLs helped prevent them from initiating smoking (Green, A. C., et al. (2014). "Investigating the effectiveness of pictorial health warnings in Mauritius: findings from the ITC Mauritius survey." Nicotine & Tobacco Research 16(9): 1240-1247). Hence, we enrolled nonsmokers to optimize the effectiveness of the HWLs for preventing WP initiation.

We added this information, along with the two references, in the “Participants and Recruitment” section to clarify our rationale:

“In addition, given the important role of the HWLs regulations in preventing smoking initiation, including WP nonsmokers in the early stage of the HWLs development ensures maximizing their benefit for WP initiation prevention (24, 25).”

8. Please provide a rationale for stratifying by gender.

Response: We stratified by gender because WP smoking prevalence, pattern, risk perception, and attitude are different by gender in the EMR and because women have some unique smoking and health concerns (e.g., WP use effects on pregnancy) (e.g. Maziak, W., et al. "Gender and smoking status-based analysis of views regarding waterpipe and cigarette smoking in Aleppo, Syria." Preventive Medicine 38.4 (2004): 479-484). We believe that paying particular attention to the gender-based differences in perceptions and attitudes related to HWLs will inform their development. This information was added in the analysis section:

“We stratified by gender because WP smoking prevalence, patterns, risk perceptions, and attitudes are different by gender in EMR, and because women have specific health concerns related to smoking effects on their offspring (30).”

9. What was the rationale for giving a presentation on the health effects of WP smoking? In practice, such additional information will not be provided when one views a warning label.

Response: The main purpose of the PowerPoint presentation of WP health effects before the discussion was to create shared baseline information that can facilitate the discussion around available evidence of WP health effects and related messages to which they will be exposed. We have added this information in the “Procedure” section in the “focus group discussion” piece. 

“The PowerPoint presentation was provided to generate basic and common knowledge about WP’s harmful effects among participants. This knowledge can facilitate the group discussion around available evidence of WP health effects and related HWLs to which they will be exposed.”

10. It is unclear what people were actually asked in the discussion. The methods talk about the framework, but how the constructs were tapped is unknown. Please provide more detail.

Response: We added this paragraph in the “Focus group discussion” section to provide more information about the questions:

“According to this model, focus group discussion focused on participants’ perceptions of each HWL in terms of (1) attention (attraction, notice, engagement, general design); (2) emotional reaction (fear, believability, avoidance); (3) effect (harm perception, addictiveness perception, intention to quit, intention not to initiate WP use); (4) recommendation for improvement (relatedness to participants, message clarity, language level and synergy with pictorials); and (5) optimal placement (size and durability of HWLs for each component - tobacco, device, charcoal) (15). The discussion guide and the baseline assessment are provided in the supplement.”

For more details, we also included the supplement's discussion guide and baseline assessment. 

11. Qualitative coding methods are rather vague. Please provide more detail on what the researchers coded for.

Response: We added to the Analysis section this paragraph to describe the coding process:

“The coding of participants was stratified based on WP use status (users, nonuser) and proceeded deductively from theoretical constructs used in the main categories (reaction, harm perception, intention to quit or not to initiate WP, and improvement). Meaning units (quotes) were grouped under their respective constructs (main categories), and summaries were drafted. A second coder reviewed the summaries and compared them to the data. Differences were resolved through peer debriefing (35). The coding of participants' recommendations for HWL changes proceeded inductively, allowing for new main categories to emerge. The constant comparative method was used to identify patterns in the data and refine the categories (36). Our frequent peer debriefing sessions included questioning each other’s interpretations of responses from several perspectives: ethnic and social backgrounds, personal histories, and whether and to what extent our characteristics, prior experiences, and knowledge influenced our interpretations of the data. Where potentially biased interpretations arose, we assessed the fitness of meaning units (quotes) to their main categorization and preliminary codes (37-39). This process was conducted iteratively as data analysis progressed.” 

Results

12. There was a different age distribution by WP smoking status (nonsmokers had many more in the two upper age categories). Any comparisons between the groups could be impacted by age. Please discuss the impact of this or if it can be controlled for.

Response: Given the qualitative nature of the study, the narrow-targeted age group (18–34 years), and that we used the same recruitment methods for both groups, we do not anticipate that the imbalance between WP smokers and non-smokers will affect our results. We addressed the reviewer’s concern by adding this information in the limitations section. 

“There was a different age distribution by WP smoking status (smokers vs. nonsmokers). However, given the qualitative nature of the study, the narrow-targeted age group (18–34 years), and that we used the same recruitment methods for both groups, we do not anticipate this imbalance affected our results.”

13. The title for Table 2 should indicate what is being presented in the cells. 

Response: We revised the title per the table content (communication outcomes). It reads now: 

“Table 2. Health warning labels rating on harm perception, intention to quit waterpipe, intention to not initiate waterpipe use, and overall effectiveness stratified by themes and waterpipe use”

14. Please edit Table 2 to include the rating scale (1 = strongly disagree, 5 = strongly agree) so that the table can be understood when it stands alone.

Response: We added a footnote in table 2 to add the requested information as follows:

“*Participants were instructed to view each label and rate it on a 5-point Likert-type scale (from 1 = strongly disagree to 5 = strongly agree) for each outcome.”

15. Overall, the tables need to be revised to improve clarity. It’s also hard to map the label numbers to what is being portrayed.

Response: We changed the table format to improve its clarity. Mainly, we (1) used the English version of the labels and increased the labels’ size and resolution, (2) added a row to specify each theme, and (3) added the full acronym to the labels (e.g., HWL12 instead of L12) to be consistent with our report in the text.

16. For Table 3, the N (%) doesn’t make sense: what is in the cell is not N (%) (unless the percentages are all less than 1%). b) The ranking % in the last column is not clear. Please better clarify what is being presented.

Response: We apologize for the confusion. We clarified that we are reporting the mean and standard deviation in the table.

We also improved the format of the table as we did in table 2 (please see response #15), and we added this information to the table footnote:

“a The overall effectiveness rating represents the mean score of four effectiveness domains including harm perception, intention to quit, preventing initiating WP use, and HWL’s overall effectiveness on other people.”

17. In order to declutter table 3 and better illustrate the ranking results, we suggested a new figure (Figure 2).

Response: Great suggestion. We added a new table (Table 4) to illustrate the ranking results separately. 

18. Please provide a better explanation of what the double asterisks footnote means in Table 3.

Response: The double asterisks means that the difference between the two comparison groups (males vs. females, or WP users vs. nonusers) is statistically significant after applying Mann Whitney test. We added this information to the table footnote:

“b p value <0.05 indicates a significant difference between the two comparison groups (males vs. females, or WP users vs. nonusers) after applying Mann Whitney test.”

19. Use of the word “theme” is assigned to both the HWL themes (1-4) and the rating categories (1-3). This is confusing. Suggest exclusively using “category” for the ratings.

Response: We reported the “Theme” in a separate row and listed only the rating category as requested. 

20. For better clarity, please make the terminology for the themes used in Figure 1 and the text in pages 13-16 identical.

Response: We reviewed the results to ensure that our terminology for the themes and HWLs is consistent in the figure, tables, and text.

21. What does this statement mean? As for the placement of HWLs, participants did not report noticeable locations.

Response: As part of the focus group discussion about the HWLs improvement, we asked participants about the best placement for the HWLs on WP components (tobacco, device, or charcoal). However, participants did not specify a preference for the HWL placement. We changed the sentence for clarification. It reads now: 

“Participants did not specify a preference for the best placement for the HWLs on WP components (tobacco, device, charcoal).” 

Discussion

22. The cited study for the following sentence is inappropriate because it is a secondary analysis of the GYTS focusing on students aged 13-15 years old. Please revise: This could be explained by the higher prevalence rate, and more intensive pattern of WP use among males compared to females in the EMR [40].

Response: We thank the reviewer for careful editing. We cited a more appropriate reference that supports our cited data:

“Asfar, T., et al., Comparison of patterns of use, beliefs, and attitudes related to waterpipe between beginning and established smokers. BMC public health, 2005. 5(1): p. 1-9.”

23. Please provide a reference to support these statements:

Males suffer from WP-related adverse health effects and dependence more than females, and therefore these warning messages are more relevant to them.

Response: We provide two references to support our statement:

• Hamadeh RR, Lee J, Abu-Rmeileh NME, Darawad M, Mostafa A, Kheirallah KA, Yusufali A, Thomas J, Salama M, Nakkash R, Salloum RG. Gender differences in waterpipe tobacco smoking among university students in four Eastern Mediterranean countries. Tob Induc Dis. 2020 Dec 2;18:100. doi: 10.18332/tid/129266. PMID: 33299390; PMCID: PMC7720794.

• Asfar, T., et al., Comparison of patterns of use, beliefs, and attitudes related to waterpipe between beginning and established smokers. BMC public health, 2005. 5(1): p. 1-9.

24. The logic of why more males rated the addictive nature of the WP label as more effective isn’t clear because the reader does not know if the participants are aware of this health disparity; in fact, the study data indicate that over half of the participants believe WP smoking is less addictive than cigarette smoking. Please refine.

Response: The misperception that WP is less addictive than cigarettes among 50% of our participants doesn’t mean that they are not addicted to nicotine. This misperception is very common. Based on robust evidence that smoking prevalence and smoking intensity/frequency of both cigarette and WP among males in the Middle East is higher than in females, it is logical to conclude that the addiction theme is more relevant to men than women, given their personal experience with addiction. 

25. The studies cited regarding overexposure of HWLs are cigarette smoking studies; thus they do not demonstrate anything for waterpipe. Since overexposure and rotation are important concepts for HWLs, please revise this language to be a recommendation in the face of what is known about cigarette HWLs.

Response: Good point. We revised the sentence to clarify that the research is related to HWL on cigarettes. It reads now:

“Indeed, several studies on cigarette HWLs demonstrated that a regular rotation and innovation of HWL design could improve its efficacy and salience (46, 47).”

26. What is meant by “advanced language”? This may be a good place to cite the Flesch Readability scale to make your point

Response: Excellent point. We apologize for using an unsuitable term. “Advanced language” refers to using complex and challenging language in the text. We added these sentences to verify this meaning. We also added the potential of using the Arabic version of the “Readability Test” to enhance the warning clarity. 

“Finally, participants recommended avoiding using complex language that might hamper the message clarity, especially among vulnerable groups such as children (56). Future research on developing new HWLs in the Middle East can benefit from using the “Arabic Readability Metric and Tool” to enhance the warning text readability (57, 58).

57. El-Haj M, Rayson PE. OSMAN: A novel Arabic readability metric. 2016.

58. Nassiri N, Lakhouaja A, Cavalli-Sforza V, editors. Modern Standard Arabic readability prediction. International Conference on Arabic Language Processing; 2017: Springer.

Reviewer 2

1. Would be useful to provide as supplemental materials the semi-structured script.

Response: We appreciate this suggestion. We added the semi-structured script as supplemental materials, as requested.

2. Any conjecture as to why smokers and nonsmokers do not differ in their evaluations. Also, it would be useful to state why we should focus on nonsmokers (e.g., to prevent uptake and susceptibility to use).

Response: Good point. The similarity in the HWLs evaluation between WP users and nonusers could be explained by the fact that WP use is widespread in the Middle East and is socially acceptable as part of the culture and history of the region. The high level of nonusers’ exposure to the habit and social interaction between users and nonusers might create a shared perception of WP’s health effects. 

Please see response # 7 for review 1 for the rationale to include nonsmokers. 

We added this information for clarification:

“However, in contrast to results from Western countries (42), no differences in participants’ reactions to the HWLs were detected based on their WP smoking status. It is important to mention here that fact that WP use is widespread in the Middle East and is socially acceptable as part of the culture and history of the region. The high level of nonusers’ exposure to the habit and social interaction between users and nonusers might create a shared perception of WP’s health effects (29, 43).

3. Describe future plans given that feedback was given as to how to improve the warnings.

Response: Excellent point. We added this paragraph to provide an insight into our future plans:

“Our next research direction is to improve the HWLs' design and content, based on our focus group study, and to further test them in a lab experiment (59). We will work with a public health designer to fine-tune the HWLs based on participants’ feedback. The top-ranked four HWLs that were selected from each theme will then be produced in high quality for testing in a lab study (59), while the developed HWLs will be used to advocate for the adoption of HWLs policies and disseminate knowledge about WP harmful effects to people in Tunisia and other Arab countries (60). In the final stage of our project, we will conduct a situation analysis of the local tobacco control policy environment in Tunisia to understand the local policy context, actors, as well as support for and barriers to HWLs policy implementation (61). The situation analysis will involve two levels: 1) analysis of official documents related to national tobacco control policy to understand policy frameworks and processes, organizational structure, and stakeholders involved in tobacco control provision and implementation, and 2) semi-structured interviews with key informants including policymakers, café owners, WP smokers/nonsmokers, civil advocates, and the media. The situational analysis results will provide a clear roadmap for effective implementation of WP HWLs.”

4. Methodologically, the warnings were obtained using the Delphi approach. This seems reasonable. However, given the large discrepancies obtained, one wonders if there is another approach that may allow investigators to match what the target audience thinks more closely. For example, one suggestion is the mental model’s approach. Perhaps this can be added in the discussion section for ways to improve the development of hookah warnings.

Response: The main purpose of our focus group study was to optimize further the HWLs developed by experts in the Delphi study by involving the target population in the HWLs development process. This methodology is highly recommended in the guidelines for developing evidence-based HWLs for tobacco products. We clarified this point more in the discussion by adding this statement:

“In contrast, these HWLs were considered important and effective by tobacco control experts in our Delphi study (19). This highlights the importance of involving the target population in creating the HWLs as recommended by the guidelines for developing evidence-based HWLs (53).”

Reviewer 3

1. Would it be better to talk about water pipe users than smokers? This was somewhat confusing to me.

Response: Good point. Both terms are used in the waterpipe literature. We changed “smokers” to “users” throughout the manuscript.

2. Is it possible to use more descriptive labels in tables 2 and 3 in order to make them more interpretable for the reader? If not, can the data be summarized and moved the tables to the appendix materials?

Response: Please see our response # 15 to Reviewer 1. We improved the two tables by increasing the size of the labels. We also used high-resolution pictures to enhance their quality. 

3. It’s not clear how the stratification of focus group participants was used for comparison purposes in the focus group analysis.

Response: The coding of participants was stratified based on WP use status (users, nonuser) Please see our response # 11 to reviewer 1, where we clarify this point.

4. Findings from the focus groups are mentioned in the discussion that are not explored in the results section. The discussion should consider the interpretation of findings rather than presenting new information. How was the theoretical model called upon in the discussion?

Responses: We listed our results and ensured that our discussion highlighted the important results. 

5. Is there a way to integrate findings from the rankings and the focus group discussion more explicitly? In the discussion, it is mentioned that the rankings were confirmed during focus group discussions, but it is not clear how this assessment was made.

Responses: Excellent point. We added this paragraph to address this comment: 

“The results of the HWLs’ ranking and rating based on overall perceived effectiveness echoed those from the focus group discussion. For example, the top-ranked HWLs (e.g., oral cancers, orally transmitted infections, and harm to children) were also perceived as the most effective during the focus group discussions, which validates and reaffirms the potential value of these HWLs in eliciting strong emotions (e.g., disgust, fear, and guilt). Similarly, HWLs that received the least effective rating (e.g. depicting CO poisoning) were also perceived as the least believable and effective during the focus group discussions. Participants indicated that these HWLs provoked poor reaction because they are didactic, illustrating scientific facts about vague chemicals (e.g., toxins) in WP or showing illnesses that are not dangerous and easy to treat, such as herpes. In contrast to this result, these HWLs were considered important and effective by tobacco control experts in our Delphi study (19), which highlights the importance of involving the target population in the development of HWLs (54).”

6. Line 386, I believe that the authors meant qualitative – but qualitative research is usually not intended to be representative, as an extrapolation from qualitative findings is not intended to be of this type.

Response: Correct. We changed the word “Quantitative” to “Qualitative.”

7. In the conclusion section, it would be good to see what messages/HWL were supported by this research.

Response: Excellent comment. We added this information in conclusions to specify the potential messages based on our results: 

“Pictorial HWLs arousing strong emotions, with visible health consequences, and depicting harm to children represent a potential target for public health WP control efforts in Tunisia. In contrast, HWLs illustrating scientific facts or exposure to chemicals without tangible harm or disease communication were perceived to be the least effective.”

---

## [Decision Letter · Decision Letter 1]

29 Nov 2022

Adapting waterpipe-specific pictorial health warning labels to the Tunisian context using a mixed method approach

PONE-D-22-10070R1

Dear Dr. Asfar,

We’re pleased to inform you that your manuscript has been judged scientifically suitable for publication and will be formally accepted for publication once it meets all outstanding technical requirements.

Kind regards,

Nabeel Al-Yateem, PhD

Academic Editor

PLOS ONE

Additional Editor Comments (optional):

Reviewers' comments:

Reviewer's Responses to Questions

**Comments to the Author**

1. If the authors have adequately addressed your comments raised in a previous round of review and you feel that this manuscript is now acceptable for publication, you may indicate that here to bypass the “Comments to the Author” section, enter your conflict of interest statement in the “Confidential to Editor” section, and submit your "Accept" recommendation.

Reviewer #1: All comments have been addressed

2. Is the manuscript technically sound, and do the data support the conclusions?

Reviewer #1: (No Response)

3. Has the statistical analysis been performed appropriately and rigorously? 

Reviewer #1: (No Response)

4. Have the authors made all data underlying the findings in their manuscript fully available?

Reviewer #1: Yes

5. Is the manuscript presented in an intelligible fashion and written in standard English?

Reviewer #1: Yes

6. Review Comments to the Author

Reviewer #1: (No Response)

7. PLOS authors have the option to publish the peer review history of their article (what does this mean?). If published, this will include your full peer review and any attached files.

Reviewer #1: No

---

## [Editor Report · Acceptance letter]

7 Dec 2022

PONE-D-22-10070R1 

Adapting waterpipe-specific pictorial health warning labels to the Tunisian context using a mixed method approach 

Dear Dr. Asfar:

I'm pleased to inform you that your manuscript has been deemed suitable for publication in PLOS ONE. Congratulations! Your manuscript is now with our production department. 

Kind regards, 

on behalf of

Dr. Nabeel Al-Yateem 

Academic Editor

PLOS ONE